# Integrated Analysis of Polycyclic Aromatic Hydrocarbons and Polychlorinated Biphenyls: A Comparison of the Effectiveness of Selected Methods on Dried Fruit Matrices

Artur Ciemniak , Agata Witczak * and Kamila Pokorska-Niewiada

Department of Toxicology, Dairy Technology and Food Storage, Faculty of Food Sciences and Fisheries, West Pomeranian University of Technology, 70-310 Szczecin, Poland
* Correspondence: agata.witczak@zut.edu.pl

**Abstract:** Polycyclic aromatic hydrocarbons (PAHs) and polychlorinated biphenyls (PCBs) are groups of chemical substances commonly found in the environment. Because of large differences in the concentrations of PAHs and PCBs in the materials tested, separate analytical methods specific to each group of compounds are usually used. The aim of this study was to compare methods for the determination of PAHs and PCBs that permit the simultaneous determination of these compounds from one solvent extract. The analysis of the content of 15 PCB congeners and 16 PAHs was conducted using dried fruits. The analyses were performed with gas chromatography coupled with mass spectrometry. PAHs and PCBs were determined separately in each fruit sample using specific extraction and cleanup procedures for the respective groups of compounds. Analyses were also performed with two methods that permitted the simultaneous analysis of PAHs and PCBs in one solvent extract. The integrated methods did not provide adequate extract cleanup of interfering substances; consequently, the results of determinations of PAHs and PCBs using these methods were significantly different from the values obtained with proven determination methods for PAHs and PCBs.

**Keywords:** polycyclic aromatic hydrocarbons; polychlorinated biphenyls; integrated analysis





## 1. Introduction

The presence of polycyclic aromatic hydrocarbons (PAHs) and polychlorinated biphenyls (PCBs) in foodstuffs raises concerns globally because they are toxic, carcinogenic, and tend to bioaccumulate in living organisms [1,2]. These compounds are classified as xenobiotics that have similar physicochemical properties, such as hydrophobicity, low degradability (especially PCBs), and lipophilicity.

According to the US Environmental Protection Agency (US EPA), 16 PAHs are classified as priority pollutants because of their prevalence and carcinogenicity [3]. In addition to carcinogens, such as benzo[a]pyrene (BaP), the EPA list includes the most common hydrocarbons, including naphthalene (Na). The EU 15 +1 PAH list was created in the European Union, and legislation focuses only on PAHs that are confirmed to be highly carcinogenic and mutagenic, regardless of their prevalence [4,5]. PAHs can damage DNA structures, cause chromosomal mutations, and increase the risk of childhood leukemia [6]. PAH content in unprocessed food derives mainly from environmental pollution, and their occurrence in fruits results from the deposition of PAH particles on plants and their absorption through the cuticle [7]. The result is the constant presence of these compounds in fruits and other edible parts of plants, which is a fundamental problem in many countries worldwide [8,9].

According to the EPA, PCBs can affect animal immune, reproductive, nervous, and endocrine systems. The regulation of all of these systems in the body is complex and

interrelated. Non-dioxin-like PCBs (ndl-PCBs) can disrupt thyroid hormone homeostasis [10] and decrease total and free thyroxine (T4) concentrations [11–13]. There are two groups of PCB compounds, the first one includes 12 congeners with similar structures and toxicological profiles to tetrachloro-dibenzo-para-dioxins (TCDD), which are known as dioxin-like PCBs (dl-PCBs). Their effects are similar to those of dioxins, including carcinogenicity, immunotoxicity, and neurotoxicity. The second group includes congeners that do not have dioxin-like properties, but which exhibit other toxic properties (ndl-PCBs) [14]. Ndl-PCB congeners have neurotoxic effects that, inter alia, inhibit tyrosine hydroxylase, an enzyme necessary for the synthesis of the neurotransmitter dopamine, or that disrupt calcium homeostasis in the nervous system [15–17]. The widespread use of PCBs in industry, especially in electrical engineering, has resulted in their widespread occurrence in aquatic and terrestrial environments. In aquatic environments, PCBs accumulate in the bodies of fishes, crustaceans, and other animals, and their bioaccumulation coefficients range from several to tens of thousands [18]. Dl-PCBs are currently recognized as carcinogenic compounds [19]. Consequently, the tolerable weekly intake (TWI) value has been lowered significantly from 14 to 2 pg-TEQ/kg bw/week [2].

Qualitative and quantitative analyses of PCBs and PAHs require many stages of sample preparation, including extracting the compounds to be tested from the matrix (e.g., food) and cleanup with large amounts of toxic organic solvents [20,21]. Thus, a more ecological, economical approach would be to develop rapid analyses that permit the simultaneous determination of many different groups of compounds. The quantitative analysis of these compounds, however, is a difficult task, because they occur in very diverse concentrations, from very low for PCBs ($ng \cdot g^{-1}$) to relatively high for PAHs ($\mu g \cdot g^{-1}$), which can result in some compounds masking the presence of others [22].

PCBs and PAHs are mainly analyzed using methods such as gas chromatography (GC) and high-performance liquid chromatography (HPLC), which are most often coupled with mass spectrometry (MS) [9,23,24]. Preparing food samples for chromatographic analysis requires the following:

- Solvent extraction using polar and non-polar solvents that are selected depending on the physicochemical properties of the compounds analyzed [25–27]. Various methods are used, including Soxhlet extraction, ultrasound-assisted extraction, microwave-assisted extraction (MAE), accelerated solvent extraction (ASE), and supercritical fluid extraction (SFE);
- Sample cleanup to eliminate lipids, sulfur, and other interfering components that hinder analyses with methods such as solid phase extraction (SPE) with silica gel, alumina, Florisil, C18 phases, amine phase adsorbents, or gel chromatography (GPC).

Determinations of chlorinated hydrocarbons (including PCBs) often include extract cleanup with sulfuric acid at one of the stages [28,29], while for PAHs, saponification is applied using alcoholic solutions of potassium hydroxide or multi-stage liquid–liquid extraction, i.e., with cyclohexane–dimethylformamide–cyclohexane [30,31].

An integrated method would minimize not only analysis time and the use of toxic solvents, but also the risk of sample contamination.

This study attempted to develop a method for simultaneously preparing samples for qualitative and quantitative analyses of PCBs and PAHs based on the methods developed by Vives and Grimalt [32] and Jaouen-Madulet et al. [33]. An attempt was also made to assess the efficiency and effectiveness of integrated methods for the simultaneous determination of PCBs and PAHs from one solvent extract and to compare them to widely used specific determination methods for individual compounds.

## 2. Materials and Methods

### 2.1. Samples and Sample Preparation

The materials used in the study were dried apricots, pears, and apples, purchased at popular retail chains in Szczecin, Poland (Table 1). Dry matter was determined using

the gravimetric method, and fat content was determined using the Soxhlet method. Lipid content was determined gravimetrically by evaporating a defined amount of the extract.

**Table 1.** Fat content and dry matter in dried fruit.

| Parameter | Dried Apricot *n* = 20 | Dried Pear *N* = 20 | Dried Apple *n* = 20 |
|---|---|---|---|
| | Mean ± SD | Mean ± SD | Mean ± SD |
| Fat content [%] | 0.25 ± 0.1 | 0.6 ± 0.1 | 1.5 ± 0.2 |
| Dry matter [%] | 72.5 ± 0.2 | 77.4 ± 0.3 | 75.2 ± 0.1 |

Note: *n*—number of samples; SD—standard deviation.

### *2.2. Chemicals*

All chemicals (cyclohexane, n-hexane, dichloromethane, acetone, sulfuric acid, Florisil, and sodium sulfate anhydrous [$Na_2SO_4$]) were of analytical grade and were purchased from Scharlau, Fluka (Germany). Deionized water was prepared using an Easy Pure UV instrument (0.05 μS/cm; Barnstead™ GenPure™ Pro, Thermo Scientific, Dubuque, IA, USA).

The dried fruit material used to test the content of compounds with each method was homogenized and freeze-dried in a LYO LAB 3000 apparatus.

The following analytical standards were used for the analyses:

- Deuterated PAHs standards (Semivolatile Internal Standard Mix) Sigma-Aldrich (Germany), naphthalene-D8 (NA D8), acenaphthene-D10 (AC D10), phenanthrene-D10 (PHE D10), chrysene-D12 (CHR D12), benzo[a]pyrene-D12 (BaP D12), and perylene-D12 (Per D12).

The standard mixture of 16 PAHs in methanol: methylene chloride (1:1) (EPA Method 610 PAH Mixture, Merc, USA) included acenaphthene (AC), acenaphthylene (ACL), anthracene (AN), benz[a]anthracene (BaA), benzo[b]fluoranthene (BbFA), benzo[k]fluoranthene (BkFA), benzo[ghi]perylene (BghiP), benzo[a]pyrene (BaP), chrysene (CHR), dibenz[a,h]anthracene (DBahA), fluoranthene (FL), fluorene (FA), indeno [1,2,3-cd]pyrene (IP), naphthalene (NA) phenanthrene (PHE), and pyrene (PY).

- Pesticides Surrogate Spike Mix solutions in acetone (4-8460, SUPELCO, Bellefonte, USA) included 2,4,5,6-Tetrachloro-m-xylene and 2,2′,3,3′,4,4′,5, 5′,6,6′-PCB.
- Standard solutions containing indicators of PCB congeners: PCB 28, PCB 52, PCB 101, PCB 138, PCB 153, PCB 180 (6 PCB–Key Isomers LGC Ltd., NE 5575, Augsburg, Taufkirchen, Germany);

dioxin-like PCBs: non-ortho PCBs (PCB 77, PCB 81, PCB 126, PCB 169) and mono-ortho PCBs (PCB 105, PCB 114, PCB 118, PCB 156, PCB 157) (CERTAN© NE 90152 LGC Ltd., Teddington, UK).

### *2.3. Extraction Procedures*

The contents of selected PAHs and PCBs in dried fruit samples were analyzed using four methods, and 10 ± 0.5 g of dried fruit was collected for each method. Methods 1 and 2 were typical, specific analytical procedures used in determinations of both groups of these compounds. Methods 3 and 4 were used to determine both groups of these compounds from one solvent extract. The following criteria were important when selecting the methods used: determination speed of selected analytes; accuracy of determinations; and economical use of chemical reagents.

### 2.3.1. Method 1—Determination of Selected PCBs

Aliquots were extracted for 8 h in a Soxhlet apparatus using 120 $cm^3$ of a 3:1 *v/v* hexane–acetone mixture. The extracts obtained were concentrated in a rotary vacuum evaporator to a volume of 2 $cm^3$ and quantitatively transferred to glass test tubes with a



capacity of 10 cm$^3$ using n-hexane. The excess solvent was evaporated in a stream of inert gas (N$_2$) to a volume of 2 cm$^3$, and the cleanup of the concentrated extracts was performed by adding oleum (7% SO$_3$ in concentrated H$_2$SO$_4$). After mixing and separating the layers, the upper layer with n-hexane and dissolved PCB compounds was collected using Pasteur pipettes and placed into new 10 cm$^3$ tubes, cleaned of acid residues by washing three times with deionized water, and dried on an anhydrous Na$_2$SO$_4$ bed (1 g).

### 2.3.2. Method 2—Determination of Selected PAHs

The aliquots were prepared, and 50 µL of PAH Semivolatile Internal Standard Mix was added to them. Then, the fat contained in the samples was saponified using 100 cm$^3$ of a 2 mol/dm$^3$ KOH in water/methanol solution (9:1 *v/v*). The samples were extracted in a Soxhlet extractor for 6 h. After the process was completed, the extracts were cooled to about 40 °C and transferred to separatory funnels with capacities of 500 cm$^3$, and 150 cm$^3$ of deionized water was added to these alkaline mixtures. Then, the PAH fractions were extracted with three portions of n-hexane (50, 30, 20 cm$^3$). The extracts were combined and again cleaned of residual potassium hydroxide (KOH), methanol, and hydrolysis products by shaking three times with 100 cm$^3$ of deionized water. If an emulsion formed, which was a common occurrence, 10 cm$^3$ of a saturated NaCl solution was added. In the next step, residual water was removed by applying the mixture to a layer of anhydrous sodium sulfate. The extracts were concentrated in a vacuum evaporator to a volume of 2 cm$^3$, followed by the SPE method. The column packing material was Florisil (1 g) roasted at 350 °C and then deactivated with a 2% addition of H$_2$O. The columns were conditioned by washing them with 6 cm$^3$ dichloromethane and then washing them twice with 6 cm$^3$ n-hexane. The test sample was transferred quantitatively to the prepared bed with 2 cm$^3$ n-hexane. The columns were washed with 10 cm$^3$ n-hexane, and the fractions containing PAH were eluted with 9 cm$^3$ of a 3:1 *v/v* n-hexane-dichloromethane mixture.

### 2.3.3. Method 3—Simultaneous Determinations of PCBs and PAHs, According to Vives and Grimalt [32]

The samples were extracted in a Soxhlet apparatus for 4 h with 150 cm$^3$ n-hexane (Figure 1). The extracts were concentrated in a rotary vacuum evaporator to a volume of 2 cm$^3$ and quantitatively transferred to 10 cm$^3$ tubes using 2 cm$^3$ isooctane. Then, the solvent was evaporated in a stream of inert gas (N$_2$) to a volume of 1 cm$^3$. The cleanup and fractionation of the extracts was performed with SPE columns filled with 5 g of alumina (Al$_2$O$_3$-activated at 120 °C). The first fraction was eluted with a 16.5 cm$^3$ n-hexane-dichloromethane mixture at a ratio of 19:1 *v/v* and then with a 3 cm$^3$ n-hexane-dichloromethane mixture at a ratio of 1:2 *v/v*. The second fraction was eluted with 13 cm$^3$ n-hexane-dichloromethane 1:2 *v/v*. In the first fraction, organochlorine compounds, including PCBs, HCB (hexachlorobenzene), and DDT (dichlorodiphenyltrichloroethane), were collected, while in the second, PAHs and HCH (hexachlorocyclohexane) were collected. The cleanup procedure for the fractions used oleum (PCBs) and columns packed with Florisil (PAHs).

### 2.3.4. Method 4—Simultaneous Determinations of PCBs and PAHs According to Jaouen-Madulet et al. [33]

The fruit samples were extracted in a Soxhlet extractor using 50 cm$^3$ of a 4:1 *v/v* n-hexane and acetone mixture. The cleanup and fractionation of the extracts obtained were performed with SPE dual-layer columns with alumina (5 g) and silica gel (5 g) that were roasted at 400 °C and then deactivated with 5% H$_2$O (Figure 1). The columns were conditioned by washing them with 20 cm$^3$ n-pentane. The prepared extract was then transferred quantitatively to a bed with n-hexane. The first fraction containing PCBs was eluted with 40 cm$^3$ n-pentane. The second fraction containing PAHs was eluted with 20 cm$^3$ of a 10:90 *v/v* dichloromethane-n-pentane mixture, followed by a 20:80 *v/v* dichloromethane-n-pentane mixture.

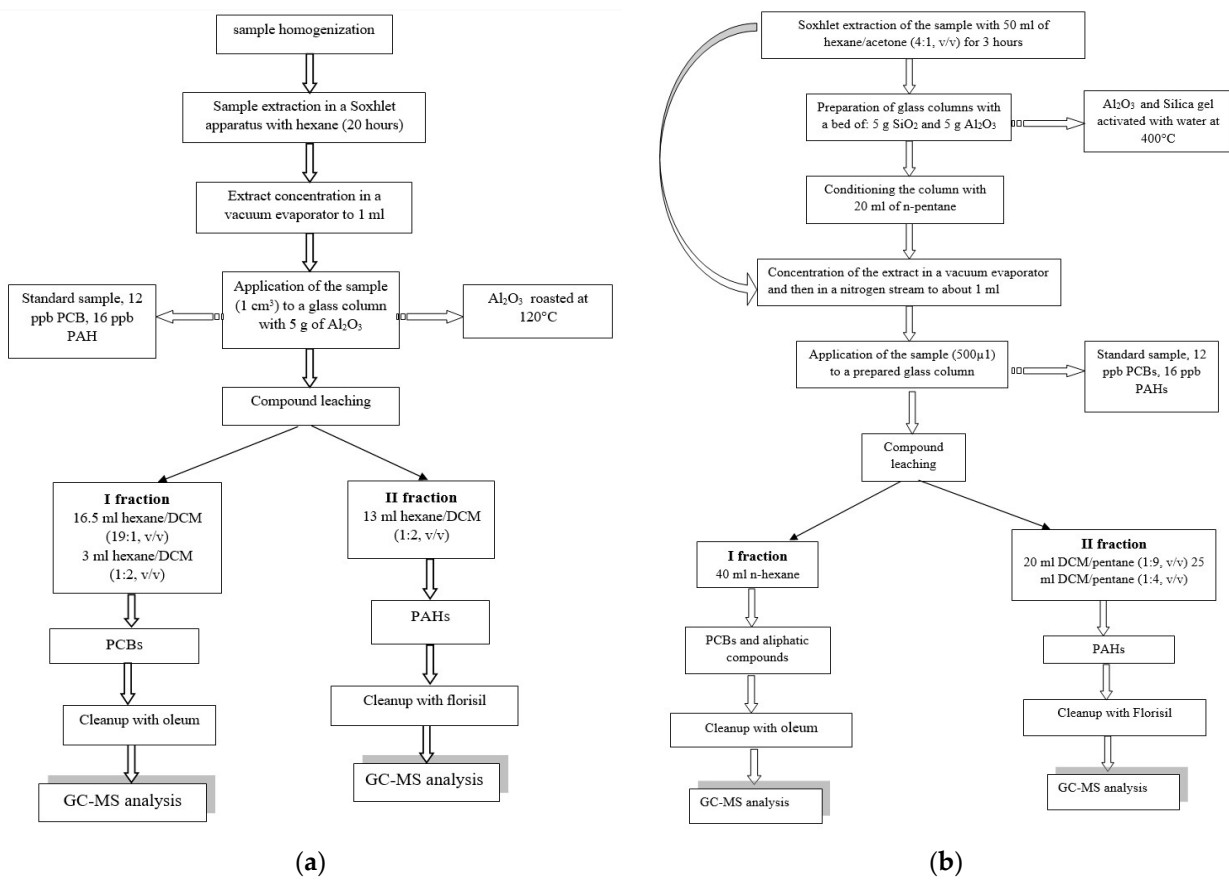

**Figure 1.** Procedures for determinations of PAHs and PCBs, according to Vives and Grimalt [32] ((**a**), method 3) and Jaouen-Madulet et al. [33] ((**b**), method 4).

All extracts obtained with all four methods were concentrated in streams of inert gas ($N_2$) to a volume of 1 cm$^3$.

### 2.4. GC-MS Analysis

The dried fruit samples were analyzed with gas chromatography coupled with mass spectrometry in a Agilent/HP 6890 GC with 5973 MSD (Palo Alto, CA, USA) in selected ion monitoring (SIM) mode. The chromatographic separation parameters are presented in Table 2.

**Table 2.** Chromatographic separation parameters.

|  | PCB | PAH |
|---|---|---|
| Sample injection | 2 μL | 2 μL |
| Column type | ZB-5MS (30 m × 0.25 μm × 250 μm) | ZB-5MS (30 m × 0.25 μm × 250 μm) |
| Carrier gas | He | He |
| Column oven temperature program | 1.1 cm$^3$/min | 1.2 cm$^3$/min |
|  | 130 °C (0.5 min *) | 80 °C (0.5 min *) |
|  | increase rate 7 °C up to 200 °C (10 min *) | increase rate 10 °C up to 230 °C (10 min *) |
|  | temperature ramp 14 °C up to 280 °C (10 min *) | temperature ramp 5 °C up to 305 °C (10 min *) |
| Electron multiplier voltage | 1920 V | 1920 V |

* thermostated.

The parameters of SIM mode determined for indicator PCB congeners and non- and mono-ortho PCBs were as follows: PCB 28—256/258 (molecular ion/confirmation ions), 186; PCB 52—256/290, 220; PCB 77—292/290, 220; PCB 81—292/290, 220; PCB 101—326/254, 328; PCB 105—326/328, 254; PCB 114—326/328, 254; PCB 118—326/328, 254; PCB 126—326/254; PCB 138—360/362, 290; PCB 153—360/290; PCB 156—360/290; PCB 157—360/290; PCB 169—360/362, 290; and PCB 180—394/396, 324.

The parameters of SIM mode during PAH analysis were as follows: NA—128/126, 102 (molecular ion/confirmation ions); ACL—152/151, 153; AC—154/153, 152; FA—166/165, 167; PHE—178/179, 176; AN—178/179, 176; FL—202/101, 203; PY—202/200, 203; BaA—228/229, 226; CHR—228/229, 226; BbF—252/253, 126; BkF—252/253, 126; BaP—252/253, 126; IP—276/138, 277; DBahA—278/139, 279; BghiP—276/138, 277; naphthalene D8—136/68, 137; acenaphthene D10—164/162, 165; phenanthrene D10—188/94, 189; chrysene D12—240/120, 241; benzo[a]pyrene D12—264/132, 265; and perylene D12—264/260, 265.

### 2.5. Analytical Method Validation for Quality Assurance

All quantifications were performed using external calibration curves. For PAH quantification, a series of solutions was prepared with the EPA 610 PAH Mix (Sigma-Aldrich, Germany). The accuracy of the analyses was verified using the internal standard method with standard solutions of deuterated aromatic hydrocarbons (Semivolatile Internal Standard Mix) [30].

The accuracy of the PCB analyses was verified by adding an internal standard using Pesticides Surrogate Spike, which is an acetone solution of decachlorobiphenyl and 2,4,5,6-tetrachloro-m-xylene. To correctly identify the PCB congeners tested, some of the samples were fortified with standard solutions of seven indicator congeners (NEN 0813) and eight toxic congeners (PCB Mix-8). A 300 µL mixture of both standard solutions, in which the concentration of each congener was 160 µg·dm$^{-3}$, was added to the samples.

Standard solutions of PAHs and PCBs were also fractionated. Each of the columns prepared for methods 3 and 4 was loaded with 1 cm$^3$ of the solutions, containing 16 PAH at 40 µg·dm$^{-3}$ each and 15 PCB at 16 µg·dm$^{-3}$ each.

The limit of detection (LOD) for each compound was determined as the concentration in the extract that produced an instrumental response at two different ions to be monitored, with a signal to noise ratio of 3:1 for the less sensitive signal [34]. A blank method was included for every 10 samples. The LOQ was estimated as 10 times the standard deviation of 10 independent blank measurements (Table 3).

**Table 3.** LOD [µg·kg$^{-1}$] and LOQ [µg·kg$^{-1}$] values for the compounds tested.

| PAH | LOD | LOQ | PCB | LOD | LOQ |
|---|---|---|---|---|---|
| NA | 0.30 | 0.91 | PCB 28 | 0.031 | 0.094 |
| ACL | 0.19 | 0.59 | PCB 52 | 0.029 | 0.087 |
| AC | 0.23 | 0.70 | PCB 101 | 0.044 | 0.132 |
| FA | 0.11 | 0.34 | PCB 81 | 0.035 | 0.110 |
| PHE | 0.08 | 0.26 | PCB 77 | 0.030 | 0.089 |
| AN | 0.11 | 0.32 | PCB 118 | 0.018 | 0.056 |
| FL | 0.05 | 0.15 | PCB 114 | 0.066 | 0.197 |
| PY | 0.05 | 0.16 | PCB 153 | 0.062 | 0.183 |
| BaA | 0.05 | 0.16 | PCB 105 | 0.060 | 0.179 |
| CHR | 0.06 | 0.18 | PCB 138 | 0.059 | 0.174 |
| BbFA | 0.08 | 0.24 | PCB 126 | 0.053 | 0.158 |
| BkFA | 0.07 | 0.22 | PCB 156 | 0.030 | 0.090 |
| BaP | 0.06 | 0.17 | PCB 157 | 0.030 | 0.091 |
| IP | 0.09 | 0.28 | PCB 180 | 0.009 | 0.028 |
| DbahA | 0.08 | 0.25 | PCB 169 | 0.040 | 0.120 |
| BghiP | 0.10 | 0.32 | | | |

Statistical analysis was performed with Statistica 13.3. The results are presented as arithmetic means. One-way ANOVA (Tukey's test; $p < 0.05$) was used to analyze the significance of the differences.

## 3. Results

The PCB and PAH contents of dried fruit were determined using four methods. Methods 1 and 2 (Figure 2) permitted determinations of PCBs (method 1) and PAHs (method 2) separately, while integrated methods 3 and 4 permitted determinations of PAHs and PCBs from single solvent extracts.

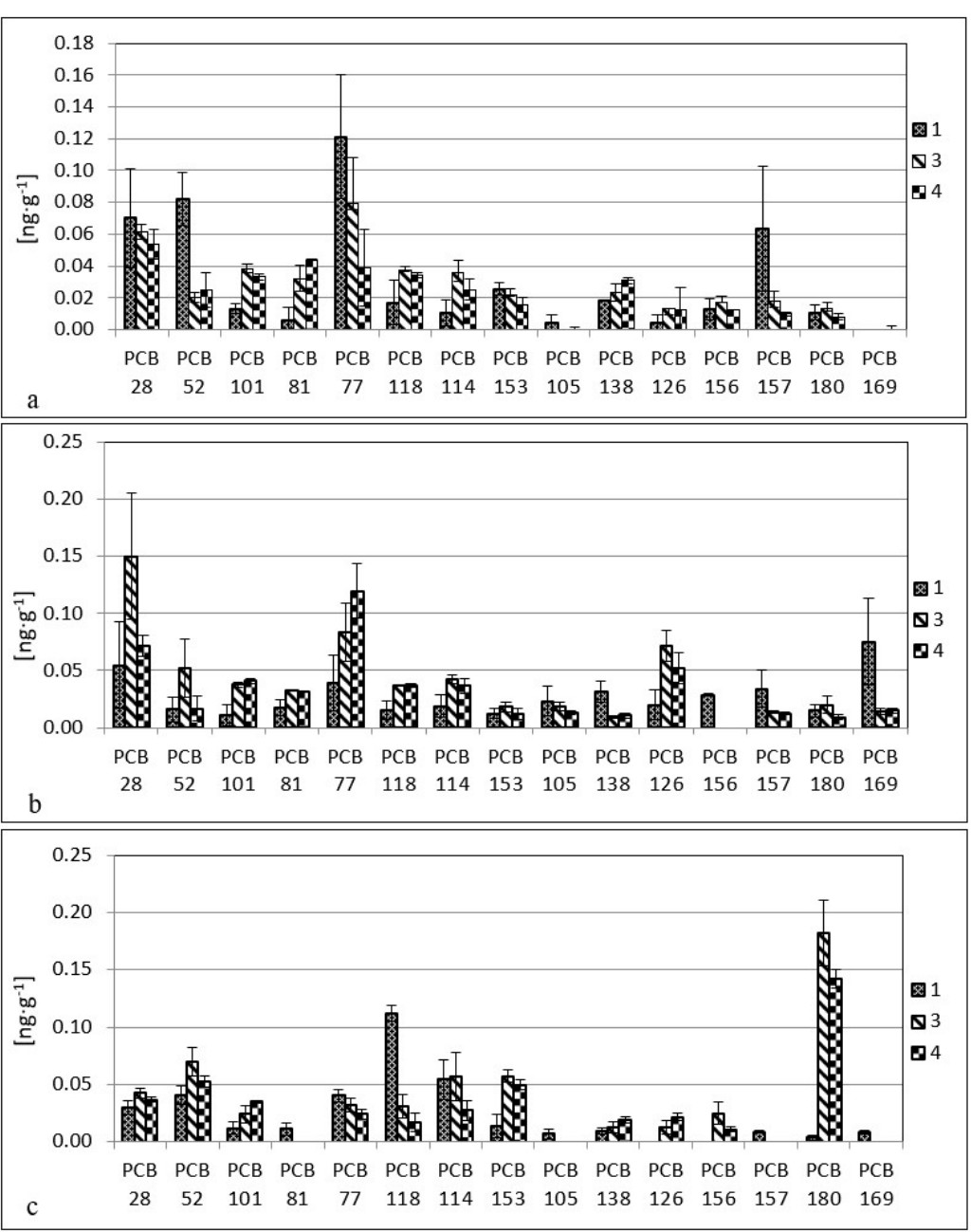

**Figure 2.** Comparison of PCB contents in dried fruits analyzed with methods 1, 3, and 4. (**a**)—apricot, (**b**)—pear, (**c**)—apple; 1—determined with the specific method for PCBs; 3—determined according to the methods of Vives and Grimalt [32]; 4—determined according to the methods of Jaouen-Madoulet et al. [33].

The values of PCB congeners in the material tested with methods 3 and 4 were significantly different from those obtained with method 1 (Figure 2, Table 4).

**Table 4.** Value ranges of PCB and PAH determinations in the fruits tested with methods 3 and 4 in comparison to method 1 (expressed in %).

| PCBs | PCB Recovery by Method 1 * (%) | | PAHs | PAH Recovery by Method 2 ** (%) | |
|---|---|---|---|---|---|
| | Method 3 | Method 4 | | Method 3 | Method 4 |
| PCB 52 | 24.7–330.9 | 30.4–127.5 | ACL | 65.7–112.7 | 55.7–211.5 |
| PCB 101 | 202.7–342.3 | 253.0–374.6 | AC | 78.4–92.7 | 63.8–233.2 |
| PCB 81 | 0–559.3 | 0.0–768.4 | FA | 71.3–83.9 | 52.7–124.3 |
| PCB 77 | 35.7–212.9 | 32.2–305.0 | PHE | 60.5–80.0 | 80.1–94.1 |
| PCB 118 | 27.6–220.3 | 14.7–233.7 | AN | 62.1–79.6 | 89.9–150.9 |
| PCB 114 | 105.4–338.3 | 50.2–238.1 | FL | 68.7–88.7 | 42.7–83.5 |
| PCB 153 | 83.9–417.3 | 60.9–361.0 | PY | 59.2–112.1 | 47.0–124.9 |
| PCB 105 | 0.0–78.2 | 0.0–55.9 | BaA | 79.6–187.9 | 39.4–129.9 |
| PCB 138 | 29.6–141.7 | 33.1–215.6 | CHR | 61.5–162.1 | 37.3–75.9 |
| PCB 126 | 330.7–358.1 | 260.6–320.8 | BbFA | 79.2–116.2 | 31.8–125.3 |
| PCB 156 | 0.0–138.3 | 0.0–100.4 | BkFA | 115.8–242.8 | 102.5–280.7 |
| PCB 157 | 0.0–39.8 | 15.7–35.2 | BaP | 70–222.9 | 61.0–513.1 |
| PCB 180 | 123.9–5284.0 | 60.9–4119.1 | IP | 45.9–480.4 | 57.9–1152.2 |
| PCB 169 | 0.0–18.5 | 0.0–19.8 | DBahA | 537.2–1323.0 | 408.1–1177.0 |
| | | | BghiP | 61.3–371.5 | 33.7–2828.7 |

* Values obtained with method 1 assumed to be 100%; ** values obtained with method 2 assumed to be 100%.

With the standard method of PCB content determination, congeners 126, 169, and 156 were not detected in the fruits, and the highest content was of PCB 77 at 0.121 ng·g$^{-1}$ dw (apricot).

The amounts of PAH compounds determined with the standard method ranged from 0.05 ng·g$^{-1}$ dw (benzo[g,h,i]terylene in apple) to 222.6 ng·g$^{-1}$ dw (phenanthrene in pear).

The analysis of PCB content using method 3 [32] did not determine the presence of congeners 105, 169, 81, 156, or 157. The highest value was for PCB 28 at 0.150 ng·g$^{-1}$ dw in pear. PAH content was determined with the same method and ranged from 0.17 ng·g$^{-1}$ dw (benzo[ghi]perylene in apple) to 179.70 ng·g$^{-1}$ dwng/g dm (phenanthrene in pear).

Method 4, developed by Jaouen-Madulet et al. [33], indicated that PCBs 105, 169, 81, 156, or 157 were not determined. The highest value determined was that of PCB 180 at 0.14 ng·g$^{-1}$ dw (apple). Determinations of PAHs indicated that the lowest content was noted for dibenzo[a,h]anthracene at 0.24 ng·g$^{-1}$ dw (apple), while the highest was for pyrene at 464.12 ng·g$^{-1}$ dw (apple).

A significant difference ($p < 0.05$) was noted for PCB congener 180 in dried apple. Values obtained with methods 3 and 4 were 54 and 42 times higher, respectively, than the value obtained with method 1. Additionally, in contrast to method 1, some PCB congeners (in apple—PCBs 81, 105, 157; in apricot—PCB 105; in pear—PCB 165) were not determined with methods 3 or 4. However, PCB congeners 126 and 156 were determined in dried apple, but were not determined with method 1, which is used specifically for determinations of PCBs (<LOD). A very low, unacceptable recovery value was noted for PCB congener 169 (Table 4).

In the current study, the model experiment conducted using standard solutions (Figure 3) indicated that methods 3 [32] and 4 [33] provided higher recovery values for all the PCB congeners analyzed, as well as for light PAHs, than did the specific methods for the two groups of these compounds. Particularly, large differences in PAH recoveries were

observed for fluorene and phenanthrene. Recovery with method 3 was 93.6–133.4% for PCBs and 18.4–94.2% for PAHs.

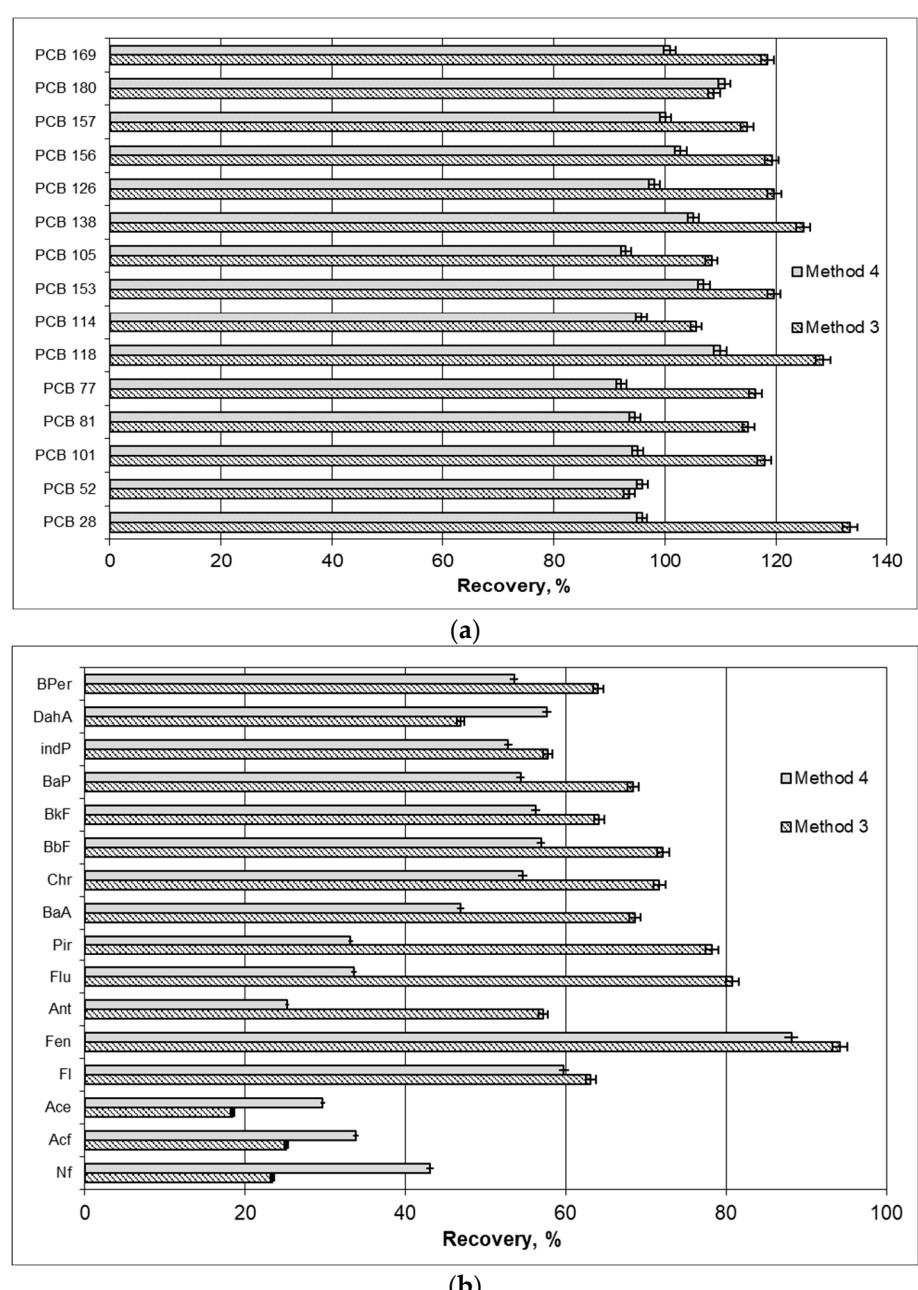

**Figure 3.** Recovery of PCBs congeners (**a**) and PAHs (**b**) (method 3—according to Vives and Grimalt [32]; method 4—according to Jaouen-Madoulet et al. [33]).

As with PCBs, the values of all 16 PAHs determined with methods 2, 3, and 4 differed depending on the method. Differences among results obtained with method 2 and methods 3 and 4 revealed that, especially with heavy PAHs, their contents were overestimated (Figure 4, Table 4). The contents of BaP determined with methods 3 and 4 were three and five times higher than that determined with method 2.

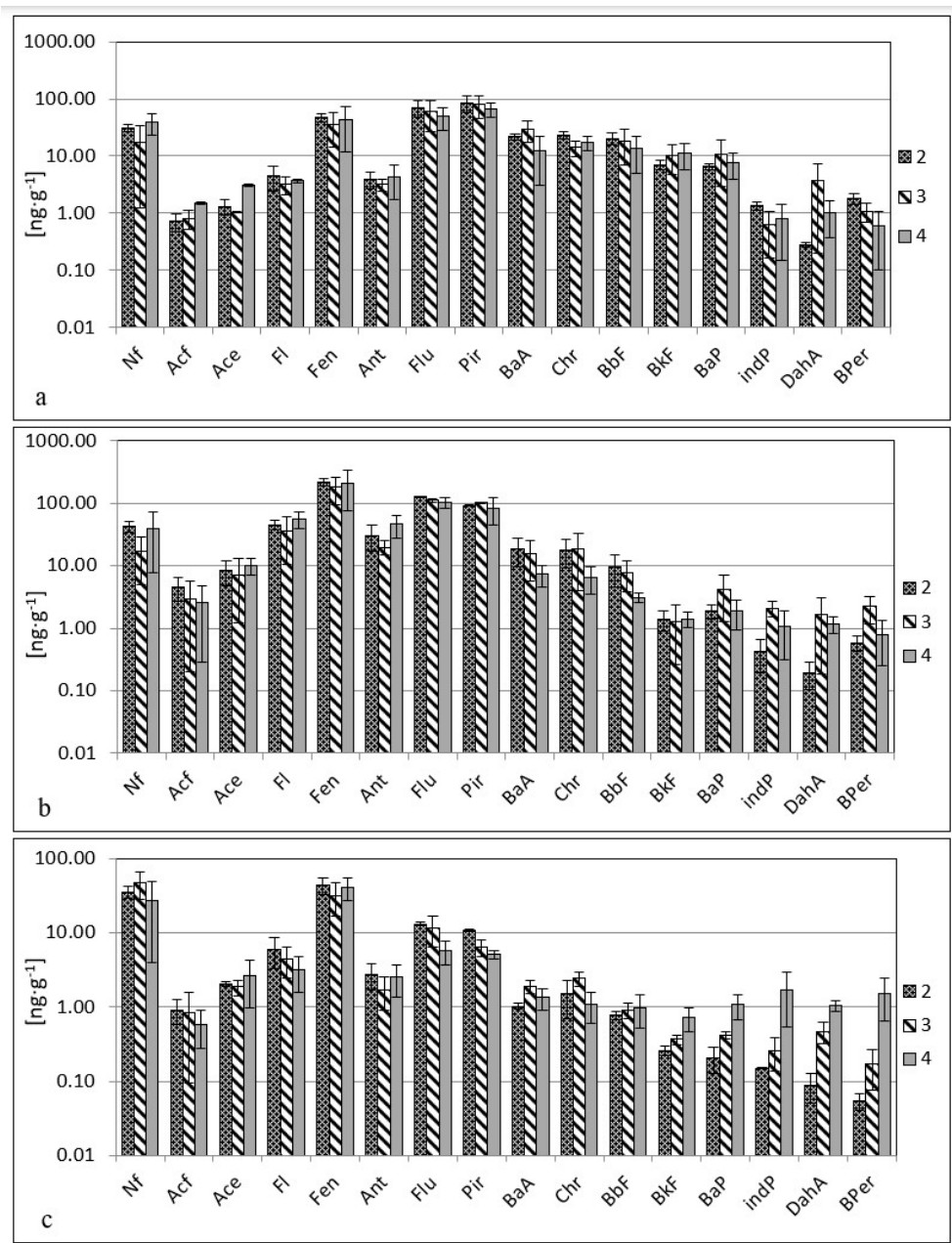

**Figure 4.** Comparison of PHA contents in dried fruits analyzed with methods 2, 3, and 4. (**a**)—apricot, (**b**)—pear, (**c**)—apple; 2—determined with the specific method for PAHs; 3—determined according to the methods of Vives and Grimalt [32]; 4—determined according to the methods of Jaouen-Madoulet et al. [33].

## 4. Discussion

Vives and Grimalt [32] developed a method for integrated determinations of PAHs, PCBs, and organochlorine pesticides in fish livers using SPE columns with alumina. The two fractions obtained are eluted with n-hexane/dichloromethane. The first extract is used for determinations of PCBs, HCB, and DDTs, while the second is used for determinations of PAHs and HCHs. The authors report that extraction in a Soxhlet extractor permitted obtaining greater analyte recovery than did saponification in an NaOH solution. The disadvantageous effects of saponification were particularly notable for chlorinated hydrocarbons, especially since the complete decomposition of HCH and pp'-DDT (dichlorodiphenyl-trichloroethane) occurred, rendering their analysis impossible. A similar method of extract

fractionation is proposed by Jaouen-Madoulet et al. [33]. Alumina and silica gel are used as the adsorbents. Individual fractions are eluted with n-pentane and a n-pentane and dichloromethane mixture. The method is used to determine PAHs and organochlorine compounds in mussels, cod liver oil, and benthic sediments. These authors, similarly to Vives and Grimalt [32], also pointed out that the procedure of analyte extraction based on saponification in an ethanolic KOH solution requires additional stages of extract cleanup due to saponification residues and results in a lower recovery values than the procedure using Soxhlet extraction. Two fractions are obtained: one containing aliphatic hydrocarbons and PCBs, and one with PAHs. Compared to the method described by Vives and Grimalt [32], this procedure requires using more solvents.

Simultaneous determinations of PAHs and PCBs in foodstuffs poses a high degree of difficulty due to the complexity of food matrices and the presence of organochlorine compounds in very low concentrations, with relatively high PAH contents. The matrix contains all the components in the sample, not just the analytes of interest. Sample extracts with high levels of organic matter, including lipids, organic acids, etc., contain substances that are co-extracted and deposited on the chromatographic column, reducing separation efficiency and interfering with the analytical method [35,36]. The matrix effect is influenced by the presence of interferences that coelute with analytes present in low concentrations [37,38]. Therefore, interference from the matrix may affect the accuracy of the analytical method. Their elimination before the final determination stage is crucial for the quality of the analysis. Therefore, it is important to use procedures to minimize the matrix effects, e.g., internal standards and the implementation of new methods of purifying the extracts. The analytical procedure that consists of the selective isolation of analytes, cleanup, and extract fractionation and quantification must also be adapted to the physicochemical properties of the compounds tested. In most studies, the contents of PAHs and organochlorine compounds in fruits are analyzed with separate methods specific to each group [20,21,39]. Relatively few researchers are currently working on developing an integrated method to analyze all of these compounds (Table 5). The methods proposed differ significantly in terms of complexity, the numbers of stages, and the sorbents used. Some commonly applied methods use, for example, Soxhlet apparatuses and sorbents that have been in use for many years [40], while others use modern sorbents and apparatuses [41–43]. Among the modern methods, zirconium oxide-based sorbents, in particular, are used instead of PSA/C18 in QuEChERS cleanup, and they ensure a lower background and higher recovery of some compounds [44,45].

Vives and Grimalt [32] reported that the PAH recovery range was 71–130%. In the current experiment, the PCB recovery range using standard solutions was 100.4–128.5%, which was higher at 79–110% than the values reported by Vives and Grimalt [32].

With method 4, the PAH recovery range reported by Jaouen-Madoulet et al. [33] was 52.8–101.4%, which was higher than the range of 30–80% obtained in the current study by using solutions. The PCB recovery range was very similar to that reported by the authors of method 4. The range was 92.17–124.9%, which was similar to that of 92.6–102.6% reported by Jaouen-Madoulet et al. [33]. Additionally, the recovery ranges for individual PAHs, regardless of the analytical method used, fluctuated more than those of the individual PCB congeners.

Additionally, inter alia, significantly elevated concentrations of heavy PAHs were noted (Figure 4). Primarily, method 2 more effectively removed impurities thanks to extraction combined with lipid alkaline hydrolysis using a methanolic KOH solution. The reasons for these discrepancies, both in the case of PAHs and PCBs, should be sought by evaluating the interference between the analyte and the compounds that were not removed during extract preparation in methods 3 and 4, i.e., fatty acids that passed into the extract. The sorbents used in this study were insufficient for simultaneously isolating PCBs and PAHs from complex organic matrices, which resulted, among other things, in overestimating the concentrations of some analytes. Effectively purifying extracts is possible using of modern sorbents. For example, it has been shown that PSA sorbents

permit the removal of all polar organic acids, polar pigments, sugars and fatty acids from extracts [46], while C18 permits the elimination of non-polar interfering substances, such as lipids (Table 5).

**Table 5.** Comparison of methods for preparing samples for determinations of PCBs and PAHs.

| Authors | Extraction Method | | Cleanup Method | | Target Compounds |
|---|---|---|---|---|---|
| Yenisoy-Karakaş and Gaga 2013 [40] | atmospheric gas, particle-phase | Soxhlet extraction: dichloromethane/petroleum ether | aluminum oxide + florisil + anhydrous sodium sulfate | | PAHs, PCBs, OCPs |
| He et al., 2017 [46] | fish | microwave-assisted extraction (MAE) | gel permeation chromatogra-phy (GPC) | neutral alumina + acid silica gel + neutral silicagel | PBDEs, PCBs |
| | | | | neutral alumina + wet neutral silica gel + wet alkaline silica gel | PBDEs, PCBs, |
| | | | | neutral alumina + wet neutral silica gel + anhydrous sodium sulfate | PAHs, OCPs |
| Lehnik-Habrink et al., 2010 [41] | forest soil | pressurized liquid extraction (PLE), acetone/cyclohexane AC/CH | GPC; silica gel or aluminum oxide | | PAH, PCBs, OCPs |
| Stenerson et al., 2016 [44] | Fatty food | QuEChERS, acetonitrile | PSA/C18, Z-Sep/C18 | | Pesticides, PAHs |
| Sapozhnikva and Lehotay 2013 [42] | fish | QuEChERS, acetonitrile | zirconia-based sorbent (Z-Sep) for d-SPE | | PCBs, PAHs, PBDEs. OCPs |
| Nácher-Mestre et al., 2014 [43] | feeds and fish tissues | QuEChERS, acetonitrile | (primary−secondary amine (PSA) + MgSO$_4$ + C18) | | Pesticides, PAHs |
| Ballesteros et al., 2009 [47] | olive oil | acetonitrile/n-hexane | GPC | | Pesticides, PAHs |
| Jaouen-Madoulet et al., 2000 [33] | environmental samples: blue mussel, cod liver oil | Soxhlet extraction: n-hexane/acetone | Alumina + silica gel | | PCBs, PAHs |
| Thompson et al., 2002 [25] | sediment | microwave-assisted extraction: dichloromethane | sulfuric acid, activated silica + activated copper | | PAHs, PCBs, OCPs |
| Vives and Grimalt 2002 [32] | fish liver | Soxhlet extraction: n-hexane/dichloromethane | aluminium oxide | | PAHs, PCBs, OCPs |
| Wolska 2002 [22] | sediment | dichloromethane | activated silica gel | | PAHs |
| | | | solvent exchanged to pentane + activated silica gel | | PCBs |

## 5. Conclusions

Determinations of PCB and PAH contents requires tedious analytical procedures. To facilitate the analyses of both groups of compounds, procedures for simultaneously preparing samples and their subsequent separation into fractions containing PCBs and PAHs are being developed.

This study compared the effectiveness of sample cleanup methods proposed by Vives and Grimalt [32] and Jaouen-Madulet et al. [33] by comparing the recoveries of the com-

pounds analyzed in dried fruit. The PCB and PAH contents obtained using the integrated determination methods were significantly different from those obtained using the specific methods for each of the groups of compounds. The integrated methods did not provide adequate extract cleanup. The extract fractionation step on alumina columns in method 3 [32] and alumina and silica gel columns in method 4 [33] were insufficient for removing compounds that interfered with the final detections. As with PCBs, the PAH contents obtained in the material tested differed depending on the methods applied.

Based on the literature, it is apparent that integrated methods for determinations of PCBs and PAHs from one solvent extract (methods 3 and 4) are less labor intensive and less time consuming, and they utilize smaller quantities of reagents. However, because the differences in individual PCB congener recoveries using methods 3 and 4 were too high compared to those in method 1, neither of these methods was accurate in testing fruit matrices, and thus, they are unsuitable for quantitative determinations of PCBs in food. The determinations of individual PAHs in methods 3 and 4 and the specific analytical method for PAHs (method 2) were consistent for light PAHs, but determinations of heavy PAHs with a higher molecular weight were overestimated in methods 3 and 4. In reference to the results of the study, sample cleanup with integrated methods 3 and 4 was satisfactory only when standard solutions were used in the model experiment. The reasons for these differences should be sought in the insufficient removal of interfering substances, mainly lipids, that were also extracted. Therefore, it is necessary to further optimize these methods or to use other sorbents.

The advantage of these methods is the possibility of performing simultaneous determinations of PCBs and PAHs, which are less labor intensive, save time, and utilize fewer reagents. However, because of inconsistent results, it is necessary to further optimize the extraction and purification stages of these methods, especially when they are applied to materials with higher fat contents.

**Author Contributions:** Conceptualization, A.C. and A.W.; methodology, A.C. and A.W.; validation, A.C., A.W. and K.P.-N.; investigations, A.C. and A.W.; original draft and manuscript preparation, A.C., A.W. and K.P.-N. All authors have read and agreed to the published version of the manuscript.

**Funding:** This research received no external funding.

**Institutional Review Board Statement:** Not applicable.

**Informed Consent Statement:** Not applicable.

**Data Availability Statement:** The data presented in this study are available upon request from the corresponding author.

**Conflicts of Interest:** The authors declare no conflict of interest.

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
