# Peer review of "Integrated Analysis of Polycyclic Aromatic Hydrocarbons and Polychlorinated Biphenyls: A Comparison of the Effectiveness of Selected Methods on Dried Fruit Matrices"

_applsci, doi:10.3390/app13064047_

Round 1

Reviewer 1 Report

I recommend that it can be accepted to publish in Applied Sciences.

Author Response

Dear Reviewer,

all authors would like to thank all the Reviewers and Editor for their comments and time spent on the manuscript evaluation. Thank you for giving us the opportunity to revise and improve our article. All the comments and suggestions from the reviewers were extremely helpful and inspiring for us, thanks to which we could improve the quality of the manuscript. All changes have been saved in change tracking mode.

The authors have made the necessary corrections according to the reviewers suggestions. The language of the manuscript has been carefully checked and corrected by a native speaker.

Thank you for your positive feedback. The manuscript has been revised as directed.

Reviewer 2 Report

You will find suggestions and comments in a separate file attached.

Author Response

Dear Reviewer,

all authors would like to thank all the Reviewers and Editor for their comments and time spent on the manuscript evaluation. Thank you for giving us the opportunity to revise and improve our article. All the comments and suggestions from the reviewers were extremely helpful and inspiring for us, thanks to which we could improve the quality of the manuscript. All changes have been saved in change tracking mode.

The authors have made the necessary corrections according to the reviewers suggestions. The language of the manuscript has been carefully checked and corrected by a native speaker.

Reviewer 2

In the abstract, the authors should omit indications of subsections such as background, methods results etc.

As recommended, the headings in the abstract have been removed

Table 1. should be transferred to the Results section.

Unfortunately, we cannot move this table, as it is mentioned for the first time in the "Materials and Methods" chapter, and according to the requirements of the journal, the table must be in the place of the first citation.

Also, I suggest figure 4. and recovery data to be transferred from the Discussion to the Results section.

Moved as recommended.

I suggest to the authors uniform the units throughout the whole manuscript (use either cm3 or ml).

Corrected

Also, better to present data as ng g-1 DW than ng/g dw. (line 285. and elsewhere).

Corrected

Introduce abbreviations before using them regardless of how well-known they are

e.g. HCB, DDT, ppDDT (lines 183-184, line 316).

Line 193- “20 cc n-pentane” supposedly should be 20 cm3?
Line 210. Indicate the model and brand of GC-MS and the country and city of the manufacturer.
Line 376. “Buy” should be by

Corrected

Would like to pose two questions to the authors as they could include their answers also in the
manuscript as a special paragraph.
1) What would be the authors’ suggestion for the integrated methods of determination of PCBs and PAHs (methods 3 and 4) how these methods could be improved in the future regarding lipid interference and what additional steps could be introduced?

The sorbents used in this study were insufficient for simultaneously isolating PCBs and PAHs from complex organic matrices, which resulted, among other things, in overestimating concentrations of some analytes. Effectively purifying extracts is possible using of modern sorbents. For example, it has been shown that PSA sorbents permit removing all polar organic acids, polar pigments, sugars and fatty acids from extracts [46], while C18 permits eliminating non-polar interfering substances such as lipids.

2) Could authors add some lines explaining the importance of the matrix effect, especially
regarding table 5. where they enlisted different references and methods used for PAHs and
PCB extraction and clean-up techniques?

The matrix contains all the components in the sample, not just the analytes of interest. Sample extracts with high levels of organic matter, including lipids, organic acids, etc., contain substances that are co-extracted and deposited on the chromatographic column, reducing separation efficiency and interfering with the analytical method. The matrix effect is influenced by the presence of interferences that coelute with analytes present in low concentrations. Therefore, interference from the matrix may affect the accuracy of the analytical method. Their elimination before the final determination stage is crucial for the quality of the analysis. Therefore, it is important to use procedures to minimize the matrix effects, e.g. internal standards and the implementation of new methods of purifing extracts.

Reviewer 3 Report

Please see the report

Author Response

Dear Reviewer,

all authors would like to thank all the Reviewers and Editor for their comments and time spent on the manuscript evaluation. Thank you for giving us the opportunity to revise and improve our article. All the comments and suggestions from the reviewers were extremely helpful and inspiring for us, thanks to which we could improve the quality of the manuscript. All changes have been saved in change tracking mode.

The authors have made the necessary corrections according to the reviewers suggestions. The language of the manuscript has been carefully checked and corrected by a native speaker.

The authors should provide a short paragraph after the third one in the introduction part
about the risk of heavy metals when they are higher than the safe limits in the human
body. So, more articles can be cited in this part such as “Utilization of PVA nano-
membrane based synthesized magnetic GO-Ni-Fe2O4 nanoparticles for removal of
heavy metals from water resources, DOI: 10.1016/j.enmm.2022.100696” and
“Development of dapsone-capped TiO2 hybrid nanocomposites and their effects on the UV radiation, mechanical, thermal properties and antibacterial activity of PVA bionanocomposites, DOI:10.1016/j.enmm.2021.100482”

 In our research, we focused solely on the analytical issue related to organic pollutants, i.e. the assessment and comparison of selected methods of integrated PCB and PAH analysis. Unfortunately we cannot add a paragraph on removing heavy metals from foods. This is too important a topic for us to mention only briefly. In addition, a publication on this topic is in preparation, so thank you for pointing out extremely interesting articles, we will use them in the future.